# eWaSR—An Embedded-Compute-Ready Maritime Obstacle Detection Network

**DOI:** 10.3390/s23125386

**Published:** 2023-06-07

**Authors:** Matija Teršek, Lojze Žust, Matej Kristan

**Affiliations:** 1Luxonis Holding Corporation, Littleton, CO 80127, USA; 2Faculty of Computer and Information Science, University of Ljubljana, Večna pot 113, 1000 Ljubljana, Slovenia

**Keywords:** maritime obstacle detection, semantic segmentation, efficient architecture, light-weight neural network, embedded hardware, OAK-D

## Abstract

Maritime obstacle detection is critical for safe navigation of autonomous surface vehicles (ASVs). While the accuracy of image-based detection methods has advanced substantially, their computational and memory requirements prohibit deployment on embedded devices. In this paper, we analyze the current best-performing maritime obstacle detection network, WaSR. Based on the analysis, we then propose replacements for the most computationally intensive stages and propose its embedded-compute-ready variant, eWaSR. In particular, the new design follows the most recent advancements of transformer-based lightweight networks. eWaSR achieves comparable detection results to state-of-the-art WaSR with only a 0.52% F1 score performance drop and outperforms other state-of-the-art embedded-ready architectures by over 9.74% in F1 score. On a standard GPU, eWaSR runs 10× faster than the original WaSR (115 FPS vs. 11 FPS). Tests on a real embedded sensor OAK-D show that, while WaSR cannot run due to memory restrictions, eWaSR runs comfortably at 5.5 FPS. This makes eWaSR the first practical embedded-compute-ready maritime obstacle detection network. The source code and trained eWaSR models are publicly available.

## 1. Introduction

Autonomous surface vehicles (ASVs) are emerging machines catering to a range of applications such as monitoring of the aquatic environment, inspection of hazardous areas, and automated search-and-rescue missions. Considering they do not require a crew, they can be substantially downsized and thus offer potentially low operating costs. Among other capabilities, reliable obstacle detection plays a crucial role in their autonomy since timely detection is required to prevent collisions which could damage the vessel or cause injuries.

The current state-of-the-art (SOTA) algorithms for maritime obstacle detection [1,2,3] are based on semantic segmentation and classify each pixel of the input image as an obstacle, water, or sky. These models excel at detecting static and dynamic obstacles of various shapes and sizes, including those not seen during training. Additionally, they provide a high degree of adaptability to challenging and dynamic water features for successful prediction of the area that is safe to navigate. However, these benefits come at a high computational cost as most of the recent SOTA semantic segmentation algorithms for maritime [1,2,3] and other [4,5] environments utilize computationally-intensive architectural components with a large number of parameters and operations. SOTA algorithms therefore typically require expensive and energy-inefficient high-end GPUs, making them unsuitable for real-world small-sized energy-constrained ASVs.

Various hardware designs have been considered to bring neural network inference to industry applications. These include edge TPUs [6], embedded GPUs [7], and dedicated hardware accelerators for neural network inference [8]. In this paper, we consider OAK-D [9], a smart stereo camera which integrates the MyriadX VPU [8]. OAK-D reserves 1.4 TOPS for neural network inference and contains 385.82 MiB on-board memory for neural networks, programs, stereo depth computation, and other image-manipulation-related operations. These properties make it an ideal embedded low-power smart sensor for autonomous systems such as ASVs.

However, state-of-the-art models, such as WaSR [1], cannot be deployed to OAK-D due to memory limitations, and many standard models that can be deployed, such as U-Net [4], typically run at less than 1 FPS, which is impractical. These limitations generally hold for other embedded devices as well. Recent works [3,10] consequently explored low-latency light-weight architectures for ASVs, but their high throughput comes at a cost of reduced accuracy. While significant research has been invested in development of general embedded-ready backbones [11,12,13,14,15,16,17,18], these have not yet been analyzed in the aquatic domain and generally also sacrifice accuracy in our experience. For these reasons there is a pressing need for ASV obstacle detection embedded-compute-ready architectures that do not substantially compromise the detection accuracy.

To address the aforementioned problems, we propose a fast and robust neural network architecture for ASV obstacle detection capable of low latency on embedded hardware with a minimal accuracy trade-off. The architecture, which is our main contribution, is inspired by the current state-of-the-art WaSR [1], hence the name embedded-compute-ready WaSR (eWaSR). By careful analysis, we identify the most computationally intensive modules in WaSR and propose computationally efficient replacements. We further reformulate WaSR in the context of transformer-based architectures and propose a Channel refinement module (CRM) and spatial refinement module (SRM) blocks for efficient extraction of semantic information from images features. On a standard GPU, eWaSR runs at 115 FPS, which is 10× faster than the original WaSR [1], with an on-par detection accuracy. We also deploy eWaSR on a real embedded device OAK-D, where it comfortably runs at 5.5 FPS, in contrast to the original WaSR, which cannot even be deployed on the embedded device. By matching WaSR detection accuracy and vastly surpassing it in speed and memory requirements, eWaSR thus simultaneously addresses maritime-specific obstacle detection challenges as well as the embedded sensor design requirements crucial for practical ASV deployment. The source code and trained eWaSR models are publicly available (https://github.com/tersekmatija/eWaSR, accessed on 25 May 2023) to facilitate further research in embedded maritime obstacle detection.

The remainder of the paper is structured as follows: Section 2 reviews the existing architectures for maritime obstacle detection, efficient encoders, and light-weight semantic segmentation architectures. Section 3 analyzes the WaSR [1] blocks, their importance, and bottlenecks. The new eWaSR is presented in Section 4 and extensively analyzed in Section 5. Finally, the conclusions and outlook are drawn in Section 6.

## 2. Related Work

### 2.1. Maritime Obstacle Detection

Classical maritime obstacle detection methods can be roughly split into detector-based methods (using Viola and Jones [19], Haar [20], or HOG [21]), segmentation-based methods (Markov random fields [22] or background subtraction [23]), and saliency map methods [24]. Bovcon and Kristan [25] use segmentation on the aligned stereo pair to estimate a single posterior over the semantic label at each pixel corresponding to the same location in the 3D scene. Muhovic et al. [26] explore 3D object detection by identifying blobs that substantially deviate from a plane fitted to the water surface point clouds obtained by stereo and use depth fingerprints for re-identification in obstacle tracking. They report failures on large featureless objects due to incorrect depth estimation. Furthermore, plane-fitting-based methods cannot detect flat objects, since these do not protrude the surface substantially. Muhovič et al. [27] improve plane-fitting using semantic image segmentation and tracking, but the method still exhibits similar depth-related limitations.

The aforementioned classical computer vision methods rely on simple hand-crafted features, which are not expressive enough for accurate detection in challenging environments. Convolutional neural networks (CNN) overcome this limitation and are widely adopted in modern methods for various tasks [1,28,29,30]. For maritime obstacle detection, several detector-based methods have been explored. Lee et al. [31] uses transfer learning on YoloV2 [32] and fine-tunes it on the SMD [33] dataset for obstacle detection. Yang et al. [34] use a deep convolutional network and a Kalman filter for high-performance object detection and tracking. Ma et al. [35] replace the backbone in Faster R-CNN [36] with ResNet [37] and improve the detection by combining multi-layer detail features and high-level semantic features with a modified DenseNet block [38].

Despite the remarkable progress of CNN-based object detection, these models cannot address static obstacles such as piers and shorelines. They also do not generalize well to previously unseen obstacles, which means that they would have to be trained on datasets that include every possible obstacle. Due to the dynamic nature of the maritime environment, the curation of such a dataset would be a costly and tedious process while still inheriting drawbacks of poor detection of unseen categories.

To address this, currently the most successful techniques adopt the principle originally proposed by Kristan et al. [22], which employ semantic segmentation to determine image regions corresponding to water (i.e., safely navigable regions) and regions corresponding to non-navigable areas (i.e., obstacles). Cane and Ferryman [39] evaluated three semantic segmentation architectures (SegNet [40], ENet [41], and ESPNet [42]) in a maritime environment and found that while the models perform well in urban scenarios, they suffer from a high false positive rate due to reflections, glare, wakes, and challenging water shapes. Bovcon et al. [43] made the same observation and proposed the first maritime obstacle detection per-pixel annotated training dataset MaSTr1325 to facilitate development of maritime-specific segmentation-based architectures. Their analysis of standard architectures (U-Net [4], PSPNet [44], and DeepLabV2 [5]) pointed out that foam, glitter, and mirroring appear to be major contributors to detection failure.

In response to the findings of [39,43], Bovcon and Kristan [1] proposed an encoder–decoder architecture WaSR based on DeepLabv2 [5] with additional blocks inspired by BiSeNet [45]. With a carefully designed decoder and a novel water-obstacle separation loss on encoder features, the model can learn dynamic water features, perform better than existing algorithms under reflections and glare and achieve state-of-the-art results. According to the major maritime obstacle detection benchmark [46], WaSR remains the current state of the art. Žust and Kristan [2] upgraded this architecture by a temporal context module that extracts spatio-temporal texture to cope with reflections. While achieving impressive results, all top-performing maritime obstacle detection methods require powerful and energy inefficient GPUs. This makes them unsuitable for small and medium-sized ASVs with limited computational and energy resources. Consequently, there is a critical need for energy-efficient detection methods applicable to embedded sensors to enhance ASVs’ operation times and energy use. Therefore, the primary motivation of our work is to develop a maritime obstacle detection method that is both computationally efficient and on par with the SOTA obstacle detection performance.

### 2.2. Efficient Neural Networks

Several works consider the general problem of efficient lightweight architectures that can run on low-power devices. GhostNet [13] applies low-cost linear operations to generate various feature maps, with the goal of revealing information of underlying intrinsic features at a lower cost. MobileNets [11,12,47] factorize convolutions into depthwise and pointwise convolutions, propose a hard swish activation function (Squeeze-and-Excite [48] blocks), and neural architecture search (NAS) to find the best architectures for mobile CPUs. ShuffleNet [14,49] introduces channel shuffling to simplify and speed-up pointwise convolutions. EfficientNets [50,51] scale the width, depth, and resolution of architectures to achieve various accuracy and latency trade-offs. RegNets [15] define convolutional network design space that provides simple and fast networks under various regimes. MicroNets [16] integrate sparse connectivity and propose novel activation functions to improve nonlinearity in low-parameter regimes. RepVGG [17] utilizes structural reparameterization to transform a multi-branch architecture during training into a plain VGG-like [52] architecture at inference time with minimal memory access. MobileOne [18] extends the idea by introducing trivial over-parameterization branches and replacing convolutions with depthwise and pointwise convolutions. The re-parametrization trick makes RepVGG and MobileOne suitable for embedded devices as they achieve good accuracy at reduced latency compared to the other state-of-the-art encoder architectures despite having more parameters and floating point operations per second (FLOPs). Metaformer [53] abstracts the transformer architecture and shows that costly attention can be replaced by a simple operation such as pooling, without substantially hampering the performance.

Several optimizations to improve and speed-up networks for the task of semantic segmentation have also been proposed. ICNet [54] uses a cascade feature/fusion unit that combines semantic information from low resolution and details from high resolution. ESPNet [42] learns the representations from a larger receptive field by using a spatial pyramid with dilated convolutions in a novel efficient spatial pyramid (ESP) convolutional module. BiSeNet [45] introduces separate spatial and semantic paths and blocks for feature fusion and channel attention to efficiently combine features from both paths in order not to compromise the spatial resolution while achieving faster inference. SwiftNet [55] triggers memory updates on parts of frames with higher inter-frame variations to compress spatio-temporal redundancy and reduce redundant computations. The recently proposed TopFormer [29] uses convolutional neural networks to extract features at different scales and efficiently extracts scale-aware semantic information with consecutive transformer blocks at low resolution. Those are later fused in a simple but fast novel semantic injection module, reducing the inference time.

## 3. WaSR Architecture Analysis

An important drawback of the current best-performing maritime obstacle detection network WaSR [1] is its computational and memory requirements, which prohibit application on low-power embedded devices. In this section, we therefore analyze the main computational blocks of WaSR in terms of resource consumption and detection accuracy. These results are the basis of the new architecture proposed in Section 4.

The WaSR architecture, summarized in Figure 1, contains three computational stages: the encoder, a feature mixer, and the decoder. The encoder is a ResNet-101 [37] backbone, while the feature mixer and decoder are composed of several information fusion and feature scaling blocks. The first fusion block is called channel attention refinement module (cARM1 and cARM2 in Figure 1) [45], which reweights the channels of input features based on the channel content. The per-channel weights are computed by averaging the input features across spatial dimensions, resulting in a 1×1 feature vector that is passed through a 1×1 convolution followed by a sigmoid activation. The second fusion block is called a feature fusion module (FFM) [45], which fuses features from different branches of the the network by concatenating them, applying a 3×3 convolution and a channel reweighting technique similar to cARM1 with 1×1 convolutions and a sigmoid activation. The third major block is called atrous spatial pyramid pooling (ASPP) [5], which applies convolutions with different dilation rates in parallel and merges the resulting representations to capture object and image context at various scales. The feature mixer and the decoder also utilize the inertial measurement unit (IMU) sensor readings in the form of a binary encoded mask that denotes horizon location at different fusion stages. In addition to the focal loss $L_{\mathrm{foc}}$ [56] for learning semantic segmentation from ground-truth labels, Bovcon and Kristan [1] proposed a novel water-obstacle separation loss $L_{R1}^{R2}$ to encourage the separation of water and obstacle pixels in the encoder’s representation space.

We note that the encoder is the major culprit in memory consumption since it employs the ResNet-101 [37] backbone. This can be trivially addressed by replacing it by any lightweight backbone. For example, replacing the backbone with ResNet-18, which uses approximately 4× fewer parameters and 18× fewer FLOPs than ResNet-101 (11.7 M and 7.2 G vs. 44.5 M and 133.8 G) and does not use dilated convolutions, thus producing smaller feature maps, already leads to a variant of WaSR that runs on an embedded device. Concretely, a WaSR variant with a ResNet-18 encoder runs at 5.2 FPS on OAK-D but suffers in detection accuracy (a 0.90% F1 drop overall and 10.79% F1 drop on close obstacles in Section 5). The performance will obviously depend on the particular backbone used and we defer the reader to Section 5, which explores various lightweight backbone replacements.

We now turn to analysis of the WaSR computational blocks in the feature mixer and decoder. The detection performance contribution of each block is first analyzed by replacing it with a 1×1 convolution and retraining the network (see Section 5 for the evaluation setup). The results in Table 1 indicate a substantial performance drop of each replacement, which means that all blocks indeed contribute to the performance and cannot be trivially avoided for speedup. Table 2 reports the computational requirements of each block in terms of the number of parameters, floating point operations (FLOPs), and the execution time of each block measured by the PyTorch Profiler (https://pytorch.org/docs/stable/profiler.html, accessed on 25 May 2023) on a modern laptop CPU. Results indicate that the FFM and FFM1 blocks are by far the most computationally intensive. The reason lies in the first convolution block at the beginning of the FFM block (Figure 1), which mixes a large number of input channels, thus entailing a substantial computational overhead. For example, the first convolution in each FFM block accounts for over 90% of the block execution time. In comparison, ASPP is significantly less computationally intensive and cARM is the least demanding block.

Both FFM and cARM blocks contain a channel re-weighting branch, which entails some computational overhead. We therefore inspect the diversity level of the computed per-channel weights as an indicator of re-weighting efficiency. The weight diversity can be quantified by per-channel standard deviations of the predicted weights across several input images. Figure 2 shows the distribution of the standard deviations computed on the MaSTr1325 [43] training set images for FFM and cARM blocks. We observe that the standard deviations for FFM blocks are closer to 0 compared to cARM1 and have a shorter right tail compared to cARM2. This suggests that the per-channel computed weights in FFM/FFM1 blocks do not vary much across the images, and thus a less computationally intensive replacements could be considered. On the other hand, this is not true for the cARM blocks, where it appears that the re-weighting changes substantially across the images. A further investigation of the cARM blocks (see information in Appendix A) shows that the blocks learn to assign a higher weight to the IMU channel in images where horizon is poorly visible. This further indicates the utility of the cARM blocks.

In terms of the WaSR computational stages, Table 2 indicates that the decoder stage entails nearly twice as much total execution time compared to the feature mixer stage. Nevertheless, since the computationally intensive blocks occur in both stages, they are both candidates for potential speedups by architectural changes.

## 4. Embedded-Compute-Ready Obstacle Detection Network eWaSR

The analysis in Section 3 identified the decoder as the most computationally and memory-hungry part of WaSR, with the second most intensive stage being the backbone. As noted, the backbone can be easily sped up by considering a lightweight drop-in replacement. However, this leads to detection accuracy reduction due to semantically impoverished backbone features. Recently, Zhang et al. [29] proposed compensating for semantic impoverishment of lightweight backbones by concatenating features at several layers and mixing them using a transformer. We follow this line of architecture design in eWaSR, shown in Figure 3.

We replace the ResNet-101 backbone in WaSR by a lightweight counterpart ResNet-18 and concatenate the features from the last layer with resized features from layers 6, 10, and 14. These features are then semantically enriched by the feature mixer stage. The recent work [29] proposed a transformer-based mixer capable of producing semantically rich features at low computational cost. However, the transformer applies token cross-attention [57], which still adds a computationally prohibitive overhead. We propose a more efficient feature mixer that draws on findings of Yu et al. [53] that computationally costly token cross-attentions in transformers can be replaced by alternative operations, as long as they implement cross-token information flow.

We thus propose a lightweight scale-aware semantic extractor (LSSE) for the eWaSR feature mixer, which is composed of two metaformer refinement modules (Figure 3)—channel refinement module (CRM) and spatial refinement module (SRM). Both modules follow the metaformer [53] structure and differ in the type of information flow implemented by the token mixer. The channel refinement module (CRM) applies the cARM [45] module to enable a global cross-channel information flow. This is followed by a spatial refinement module (SRM), which applies sARM [58] to enable cross-channel spatial information flow. To make the LSSE suitable for our target hardware, we replace the commonly used GeLU and layer normalization blocks of metaformer by ReLU and batch normalization. The proposed LSSE is much more computationally efficent than SSE [29]. For example, with a ResNet-18 encoder, the SSE would contain 66.4 M parameters (requires 3.2 GFlops), while the LSSE contains 47.9 M parameters (requires 2.1 GFlops).

We also simplify the WaSR decoder following the TopFormer [29] semantic-enrichment routines to avoid the computationally costly FFM and ASPP modules. In particular, the output features of the LSSE module are gradually upsampled and fused with the intermediate backbone features using the semantic injection modules [29] (SIM). To better cope with the high visual diversity of the maritime environment and small objects, the intermediate backbone features on the penultimate SIM connection are processed by two SRM blocks. The final per-layer semantically enriched features are concatenated with the IMU mask and processed by a shallow prediction head to predict the final segmentation mask. The prediction head is composed of a 1×1 convolutional block, a batchnorm and ReLU, followed by 1×1 convolutional block and softmax.

## 5. Results

### 5.1. Implementation Details

eWaSR employs the same losses as WaSR [1]: Focal loss [56] is applied on segmentation along with L${}_{2}$ weight decay regularizer [59] and the water-obstacle separation loss [1] is applied to the backbone features (layer 14 in ResNet-18) to encourage learning embedding with well separated water and obstacle features. Network training follows the procedure in [1]. We train for 100 epochs with the patience of 20 epochs on validation loss, RMSProp optimizer with 0.9 momentum, and learning rates of $10^{-6}$ and $10^{-5}$ for backbone and decoder, respectively. Contrary to WaSR [1], we train eWaSR and other networks that rely on metaformers with a batch size 16 and a maximum of 200 epochs since transformers benefit from longer training on larger batches [60]. We refer the reader to our official eWaSR implementation (https://github.com/tersekmatija/eWaSR, accessed on 25 May 2023) for more implementation details and pre-trained networks for results reproduction.

### 5.2. Training and Evaluation Hardware

All networks are trained on a single NVIDIA A4000 GPU. A laptop GPU NVIDIA RTX 3070Ti is used for GPU latency estimation, while the on-device performance tests are carried out on the embedded OAK-D [9] with 16 SHAVE cores, delivering 1.4 TOPS for on-device neural network inference. The power consumption of OAK-D is approximately 7.5 W, which is far below 290 W of the thermal design power (TDP) typical for user-grade GPUs such as Nvidia RTX 3070Ti, thus making it a suitable embedded device testing environment.

While OAK-D executes neural networks on-chip, it still requires a host machine (such as laptop or Raspberry Pi) that loads the pipeline with the models to the device. Because it is not possible to measure the raw inference time of the neural network execution, we measure the latency from the image sensor to receiving the output on the host computer. We reason that this is sufficient since this mimics the use of OAK-D in potential deployment, with inference being performed on OAK-D and the rest of navigation logic on the host machine. To reduce the effect of parallel execution, we allow only one image to be in the pipeline’s queue and use a single neural network processing thread when measuring the latency. The latency is thus estimated as the time between the outputs, averaged over 200 outputs. We use both inference threads when reporting FPS.

### 5.3. Datasets

The evaluation follows the established performance evaluation protocol from the maritime obstacle detection benchmark MODS [46]. All architectures are trained on the Maritime Semantic Segmentation Training Dataset MaSTr1325 [43] and evaluated on the MODS test set. MaSTr1325 contains 1325 per-pixel annotated images captured by an ASV. The images contain various obstacles of different sizes and shapes, such as swimmers, buoys, seagulls, ships, and piers, and are captured in different weather conditions. The pixels are annotated as either obstacle, sky, water, or unknown/ignore class. Each image is accompanied by a binary IMU mask, which is obtained by transforming the IMU readings into a horizon position in the image and setting all pixels above the horizon to 1 and the rest to zero.

To estimate the readiness of the models for the deployment to real-life applications, we benchmark them on the currently most challenging maritime obstacle detection benchmark MODS [46] since it covers a wide range of conditions, including occlusion, illumination, and other image-related challenges that affect real-world detection performance on ASVs. MODS consists of 94 sequences with approximately 8000 annotated images collected with an ASV. Each frame is accompanied by the IMU mask. Unlike MaSTr1325, MODS does not include per-pixel labels and is instead designed with simpler annotations to evaluate models in two important aspects for ASV navigation—water-edge estimation and obstacle detection. Water-edge estimation measures the accuracy of water–land boundary predictions as a root mean squared error (RMSE) between the predicted water boundary and the ground truth annotations (water-edge polyline). Ground-truth (GT) dynamic obstacles are annotated as bounding boxes. To measure the obstacle detection performance, a prediction is considered as a true positive (TP) if a sufficient area of the GT bounding box is covered with an obstacle mask. If this condition is not met, the GT obstacle is considered a false negative (FN). Blobs predicted as obstacles in the water regions outside the GT bounding boxes are considered false positives (FP). The obstacle detection task is finally summarized by precision $\left(\mathrm{Pr}=\frac{\mathrm{TP}}{\mathrm{TP}+\mathrm{FP}}\right)$, recall (Re=TPTP+FN), and F1 (=2×Pr×RePr+Re) score and is evaluated for all obstacles in the image (reported as Overall) and obstacles at the maximum distance of 15 meters from the vessel (reported as Danger Zone), which pose an immediate threat to the vehicle. In this paper, we report the performance of both aspects of MODS but put more focus on the obstacle detection aspect as it is more important for the safe navigation of ASVs. Examples of images from the MaSTr1325 and MODS datasets are shown in Figure 4.

### 5.4. Influence of Lightweight Backbones on WaSR Performance

As noted in Section 3, WaSR [1] can be trivially modified to run on embedded device by replacing its Reset-101 backbone with a lightweight counterpart. We explore here several alternatives to arrive at a strong baseline. In particular, the following eight lightweight backbones are considered: ResNet-18 [37], RepVGG-A0 [17], MobileOne-S0 [18], MobileNetV2 [12], GhostNet [13], MicroNet [16], RegNetX [15], and ShuffleNet [49]. The backbone implementations are based on official ([13,16,17,18]) and torchvision [61] implementations.

Table 3 reports the obstacle detection performance, the inference times on GPU and OAK-D, and the combined number of channels in the intermediate multi-scale features of the encoder for several lightweight backbones. The WaSR variant with a ResNet-18 [37] backbone achieves the highest F1 score overall (92.56%), while the model with a RegNetX [15] encoder performs the best inside the danger zone (83.95% F1) at the cost of being the slowest on OAK-D (395.22 ms). In contrast, MicroNet-M0 is the fastest on OAK-D (75.0 ms) but achieves the lowest overall F1 score (68.49%). GhostNet [13] is the fastest on the GPU (5.55 ms). In addition to GhostNet, the latency of MicroNet-M0 [16], GhostNet, and MobileNetV2 [12] is also below 150 ms and 6 ms on OAK-D and GPU, respectively. All three models have less than 472 total channels, followed by ResNet-18 encoder with 960 channels. Despite having a larger total number of channels from the intermediate features, RepVGG [17] and MobileOne [18] achieve a similar overall (91.81% and 91.25%) and better danger zone F1 scores (80.73% and 82.41%) than the ResNet-18 model (92.56% overall F1, 78.09% danger zone F1), with a lower latency (225.02 and 283.89 ms, respectively, compared to 314.64 ms). The variant of WaSR with a ResNet-18 backbone achieves the best overall performance at low latency, which justifies it as a strong embedded-ready baseline and is referred to as *WaSR-Light* in the following analysis.

### 5.5. Comparison with the State of the Art

We first compare our eWaSR with the baseline model WaSR-Light and the original WaSR [1]. Results are reported in Table 4. eWaSR most substantially outperforms WaSR-light inside the danger zone (+9.54% F1), which is the most critical for safe navigation. eWaSR is approximately 10% slower on GPU compared to the WaSR-Light (8.70 ms compared to 7.85 ms, respectively), but is approximately 5% faster on the embedded hardware (300.37 ms compared to 314.64 ms on OAK-D, respectively). The F1 performance of eWaSR is only slightly reduced compared to WaSR (−0.52% and −0.73% overall and inside the danger zone). Thus the detection performance of eWaSR is comparable with the original WaSR while being over 10× faster on GPU (8.70 versus 90.91 milliseconds). Additionally, in contrast to WaSR, which cannot even be deployed on OAK-D due to memory constraints, eWaSR can comfortably run on-device at 5.45 FPS.

eWaSR is compared with 12 state-of-the-art or high-performing architectures designed for maritime or general semantic segmentation. Specifically, we evaluate the performance of BiSeNetV1${}_{\mathrm{MBNV}2}$ [12,45], BiSeNetV2 [62], DDRNet23-Slim [63,64], EDANet [65], EdgeSegNet [66], LEDNet [67], MobileUNet [68], RegSeg [69], ShorelineNet [3], DeepLabV3${}_{\mathrm{MBNV}2}$ [5,12], ENet [41], and Full-BN [10]. Table 4 shows that eWaSR outperforms all these methods in the overall and danger-zone F1 score. It ranks fifth in latency on OAK-D but outperforms all faster models by over 9.74% and 22.7% F1 overall and inside the danger zone, respectively. The excellent latency–performance trade-off of eWaSR is further confirmed in Figure 5.

Figure 6 shows qualitative comparison of eWaSR, WaSR-Light, and WaSR for further insights. We observe that eWaSR predicts fewer false positives in the presence of land and in the distance and predicts more accurate segmentation masks on distant vessels and their reflections. However, its performance is limited on large homogeneous and textureless surfaces where it is prone to misclassification. On the contrary, WaSR performs better in such cases, but predicts more false positives in presence of the piers. All models appear to experience false detections in some cases in the presence of wakes and glares, which is a point requiring further research in maritime obstacle detection.

### 5.6. Ablation Studies

#### 5.6.1. Influence of Backbones

As in the case of WaSR-Light, we analyze the impact of different lightweight backbones on eWaSR performance. Based on results from Table 3, we consider the backbones that do not reduce performance or incur a minimal reduction but also reduce the latency: RepVGG-A0 [17], MobileOne-S0 [18], RegNetX [15], MobileNetV2 [12] and GhostNet [13].

Results in Table 5 show that the fastest processing times are achieved by using GhostNet (OAK-D 174.97 ms) and MobileNetV2 (GPU 8.25 ms). However, this comes at a substantial danger-zone F1 score reduction compared to ResNet-18 (−17.88% and −8.48%, respectively). The latter achieves the best overall and danger-zone F1 scores (93.02% and 87.63%, respectively) while still maintaining a good latency (300.37 ms on OAK-D). Contrary to the results in Table 3, MobileOne-S0 and RepVGG-A0 encoders produce a higher latency compared to eWaSR (388.44 ms and 374.95 ms, respectively) due to a high number of channels in the intermediate features (in total 1616, 1456, respectively, compared to 960 from ResNet-18). Consequently, FLOPs of eWaSR’s lightweight scale-aware semantic extractor (LSSE) are 2.8× and 2.3× higher compared to LSSE in eWaSR with ResNet-18 encoder at no F1 improvement. This shows that while both RepVGG-A0 and MobileOne-S0 reduce the latency at minimal accuracy cost in classification networks, they are not suitable for eWaSR or similar architectures operating on concatenated intermediate features. While eWaSR is not the fastest with ResNet-18 encoder, it outperforms other combinations by at least 0.17% overall F1 and 1.02% danger-zone F1 while still maintaining a good latency (300.37 ms and 8.70 ms on OAK-D and GPU, respectively). Thus, ResNet-18 is suggested as the primary backbone of eWaSR.

#### 5.6.2. Token Mixer Analysis

To better understand the effect of spatial refinement in form of SRM for maritime obstacle detection in eWaSR, we perform two ablation studies. Table 6 shows that removing the proposed SRM on the penultimate SIM connection (Figure 3) results in a minimal overall performance drop (−0.08% F1), but the drop is particularly apparent within the danger zone (−3.87% F1). Note that the removal does not substantially improve the inference time (−0.13 ms). The proposed spatial refinement module is thus crucial for improved detection accuracy of small objects yet does not bring substantial computational overhead.

Next, we turn to the lightweight scale-aware semantic extractor block, which also contains two SRM blocks. Table 6 shows that replacing these two with cARM-based CRM blocks results in a slight overall performance drop (−0.09% F1 compared to eWaSR), but a substantial drop inside the danger zone (−2.89% compared to eWaSR). While the latency is comparable, eWaSR works slightly faster than the variant with CRMs (−0.35 ms), since the convolutions in SRM operate on two channels only. The results thus confirm the performance benefits of SRM’s spatial refinement utilizing sARM in eWaSR.

#### 5.6.3. Channel Reduction Speedup

As discussed in the beginning of the subsection, the results of Table 5 showed that the higher number of feature channels on skip connections connected to the decoder blocks increase the latency of eWaSR. We thus explore potential speedups by projection-based channel reduction before concatenation of the encoder features into the LSSE and decoder, which maps all features to a lower dimension.

For easier comparison, we only focus on eWaSR with ResNet-18 encoder. Using 1×1 convolution, we halve the number of channels in each intermediate encoder output. As a result, the complexity of LSSE is noticeably decreased. In the case of ResNet-18 encoder, concatenated features only have 480 channels compared to 960 without projection. In Table 6, we compare the results of the model with projected features to eWaSR.

The model with projected features achieves slightly lower F1 scores (−0.63% overall F1 and 1.82% danger-zone F1, respectively) at approximately 13% speed-up (38.14 ms faster) on OAK-D, which means the projection trick is a viable option when one aims for a faster rather than a more accurate model and can be used in practical setups when a specific trade-off is sought in eWaSR performance.

## 6. Conclusions

We presented a novel semantic segmentation architecture for maritime obstacle detection eWaSR, suitable for deployment on embedded devices. eWaSR semantically enriches downsampled features from ResNet-18 [37] encoder in a SSE-inspired [29] lightweight scale-aware semantic extraction module (LSSE). We propose transformer-like blocks CRM and SRM, which utilize cARM [45] (channel-attention) and sARM [58] (simplified 2D spatial attention) blocks as token mixers, instead of costly transformer attention, and allow LSSE to efficiently produce semantically enriched features. Encoder features are fused with semantically enriched features in SIM [29] blocks. To help the model extract semantic information from a more detailed feature map, we use two SRM blocks on the second long skip connection, and we concatenate binary encoded IMU mask into the prediction head to inject information about tilt of the vehicle.

The proposed eWaSR is 10× faster than state-of-the-art WaSR [1] on a modern laptop GPU (8.70 ms compared to 90.91 ms latency, respectively) and can run comfortably at 5 FPS on embedded device OAK-D. Compared to other lightweight architectures for maritime obstacle detection [3,10], eWaSR does not sacrifice the detection performance to achieve the reduced latency and achieves only 0.52% worse overall and 0.73% danger-zone F1 score on the challenging MODS [46] benchmark compared to state of the art.

Because of additional memory access, long-skip connections can increase the overall latency of the network. In the future, more emphasis could be put on exploring different embedded-compute-suitable means of injecting detail-rich information to the decoder. The developed embedded-compute-ready architecture can be further extended in several ways. One example is to consider the temporal component as in [2]. Furthermore, since OAK-D is capable of on-board depth computation, fusion of depth into the model could be explored to increase the performance on close obstacles. Alternatively, the solution could be redesigned into a multitask architecture that would also provide an estimate of the distance besides existing semantic segmentation output. Potential practical applications (for example autonomous ferries [70], collision avoidance systems [71]) could easily combine predicted segmentation masks with depth, which would provide ASVs with precise information about distance and location of the nearby obstacles. We delegate these improvements and implementation of a practical application to future work.

## Figures and Tables

**Figure 1 sensors-23-05386-f001:**
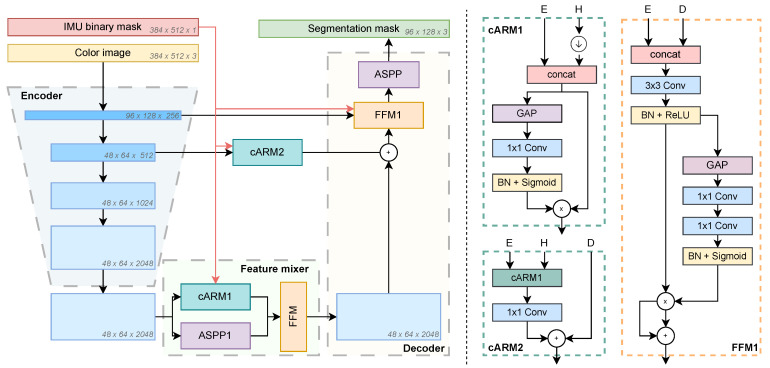
The WaSR [1] architecture (**left**) and the feature fusion (FFM1) and channel attention refinement (cARM1 and cARM2) blocks (**right**). The FFM block on the left is same as FFM1, except it takes as the input the encoder features, the downsampled IMU mask, and the upsampled decoder features. The symbol ↓ stands for downsampling, E stands for encoder features, D for decoder features, and H for binary IMU mask (horizon) features. The shapes of inputs, backbone features, and output are denoted in italics.

**Figure 2 sensors-23-05386-f002:**
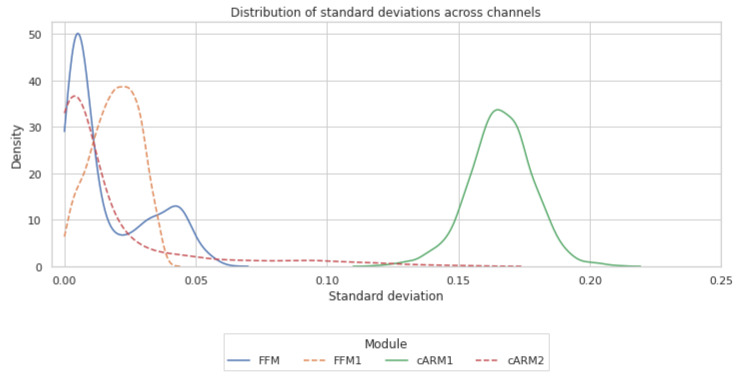
Distribution of per-channel standard deviations of channel re-weighting sigmoid activations across images of the MaSTr1325 training set in the cARM and FFM blocks. Lower standard deviation indicates that channels are given a similar weight in each image.

**Figure 3 sensors-23-05386-f003:**
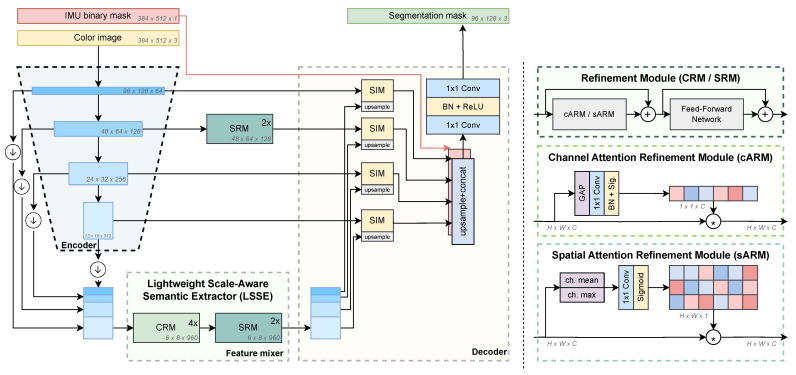
The eWaSR architecture follows the encoder, feature mixer, and decoder architecture. The backbone features of intermediate encoder layers are resized, concatenated, and processed by a lightweight scale-aware semantic extraction module (LSSE) consisting of channel refinement (CRM) and spatial refinement modules (SRM) visualized on the right. The semantically-enriched features are injected into higher-layer backbone features by semantic-injection modules [29] (SIM). The resulting features are then concatenated with the IMU mask and passed to the segmentation head. We denote shapes of inputs, backbone features, and output in italics.

**Figure 4 sensors-23-05386-f004:**
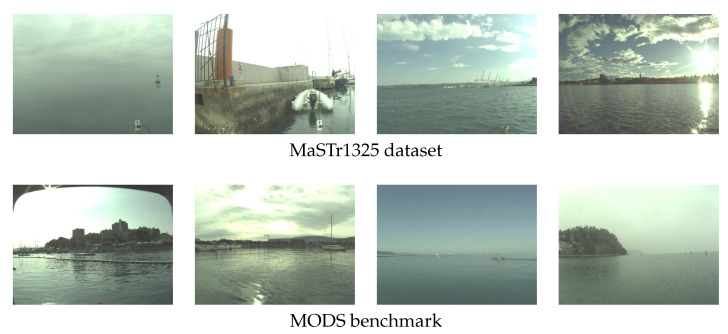
Example images from the MaSTr1325 [43] dataset used for training and the MODS [46] test dataset used for evaluation. Both datasets include challenging scenarios with varied obstacles and diverse weather conditions.

**Figure 5 sensors-23-05386-f005:**
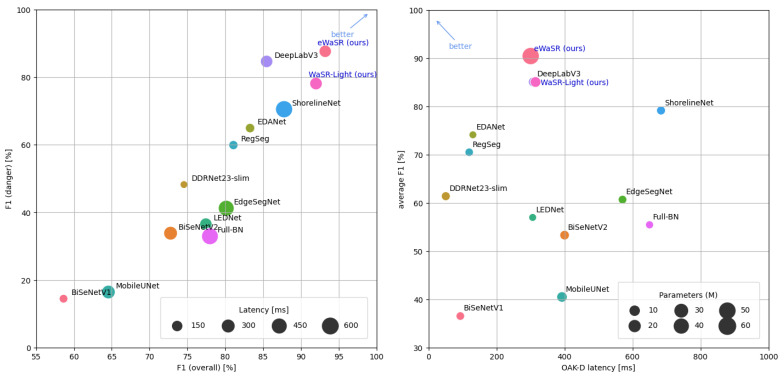
The latency–performance trade-offs of the tested methods. We visualize the average of overall and danger zone F1 in the right plot. eWaSR achieves the best overall and danger zone F1 score at a low latency on OAK-D.

**Figure 6 sensors-23-05386-f006:**
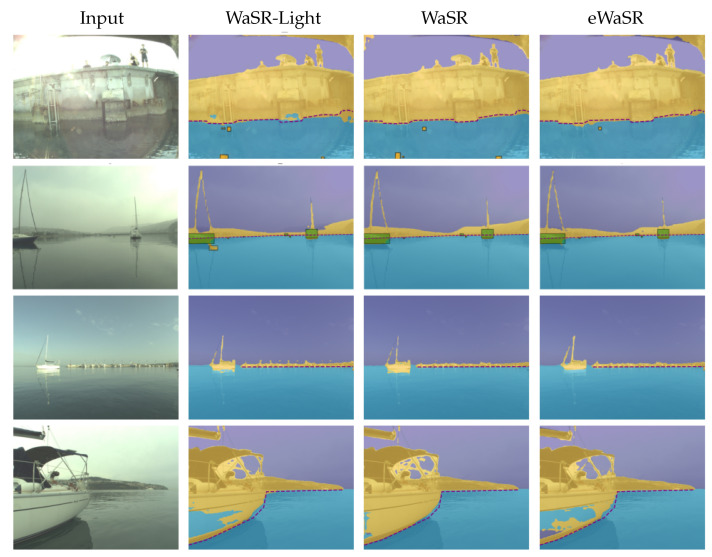
Performance of eWaSR, WaSR-Light, and WaSR on the MODS benchmark [46]. In the first column, we show an input image, followed by results of the baseline model, original WaSR model, and the proposed eWaSR. We can see that eWaSR has fewer FP predictions when close to piers (**1st row**) or in distance (**2nd row**). It can predict more refined segmentation masks than baseline in the distance (**3rd row**) but similarly struggles on flat surfaces of boats (**4th row**). We show TP (green) and FP (orange) bounding boxes in the first two rows but omit them in the last two for better comparison.

**Table 1 sensors-23-05386-t001:** The overall F1 score on MODS [46] for WaSR with individual modules replaced by 1×1 convolution (indicated by “−”). The blocks corresponding to the feature mixer are denoted by light gray.

WaSR	Original	−ASPP1	−cARM1	−FFM	−cARM2	−ASPP	−FFM1
F1	93.5%	−0.8%	−0.8%	−1.2%	−1.0%	−1.4%	−0.7%

**Table 2 sensors-23-05386-t002:** WaSR decoder computational analysis in terms of number of parameters (in millions), FLOPs (in billions), and the PyTorch profiler output on the CPU. We report the total execution time of each block as well as the execution of the slowest operation in each block. The blocks corresponding to feature mixer are denoted by light gray. We denote the metrics for the two slowest modules (FFM and FFM1) in bold.

Block	Parameters (M)	FLOPs (B)	Total Execution Time [ms]	Slowest Operation [ms]
ASPP1	1.77	5.436	48.83	18.06
cARM1	4.20	0.0105	12.36	3.99
FFM	**21.28**	**58.928**	**279.40**	**266.32**
cARM2	0.79	1.617	19.81	10.54
ASPP	0.11	1.359	66.75	17.16
FFM1	**13.91**	**145.121**	**641.23**	**589.55**

**Table 3 sensors-23-05386-t003:** Lightweight WaSR variants with different backbones comparison in terms of water-edge accuracy, overall and danger zone F1 score, the latency on OAK-D and GPU, and the total number of channels of the multi-scale encoder. Gray highlight denotes our chosen baseline model WaSR-Light and the best results for each metric are denoted in bold.

		Overall			Danger Zone (<15 m)			Latency		
Encoder	W-E	Pr	Re	F1	Pr	Re	F1	OAK	GPU	chs
ResNet-18 [37]	20 px	93.46	91.67	92.56	66.98	93.61	78.09	314.64	7.66	960
RepVGG-A0 [17]	19 px	91.71	91.9	91.81	70.49	94.44	80.73	225.02	6.06	1616
MobileOne-S0 [18]	18 px	92.2	90.33	91.25	73.59	93.64	82.41	283.89	6.47	1456
MobileNetV2 [12]	24 px	90.05	85.99	87.98	63.71	91.69	75.18	149.97	5.66	472
GhostNet [13]	21 px	90.47	89.72	90.1	59.4	92.96	72.49	129.81	5.55	304
MicroNet [16]	43 px	63.59	74.22	68.49	15.28	71.8	25.19	75.0	5.60	436
RegNetX [15]	18 px	91.89	89.55	90.7	76.75	92.65	83.95	395.22	9.80	1152
ShuffleNet [49]	23 px	90.14	87.38	88.74	61.18	90.95	73.15	274.93	8.10	1192

**Table 4 sensors-23-05386-t004:** Comparison of the proposed eWaSR (blue highlight) with state-of-the-art segmentation models and WaSR-Light (gray highlight). Best results are shown in bold, while (/) indicates that the model cannot be deployed on OAK-D due to unsupported operations or memory constraints. The best results for each metric are denoted in bold.

		Overall			Danger Zone (<15 m)			Latency	
Model	W-E	Pr	Re	F1	Pr	Re	F1	OAK	GPU
BiSeNetV1_MBNV2 [12]_ [45]	46 px	45.94	80.94	58.62	7.9	83.32	14.44	93.45	3.52
BiSeNetV2 [62]	36 px	64.96	82.71	72.77	21.55	78.54	33.82	400.08	4.46
DDRNet23-Slim [63,64]	54 px	74.82	74.24	74.53	35.5	75.08	48.21	50.70	4.71
EDANet [65]	34 px	82.06	84.53	83.28	52.82	84.22	64.92	130.45	6.09
EdgeSegNet [66]	58 px	75.49	85.39	80.13	27.48	82.33	41.21	570.52	11.36
LEDNet [67]	92 px	74.64	80.46	77.44	24.33	73.01	36.5	306.25	9.48
MobileUNet [68]	47 px	52.54	83.68	64.55	9.21	75.36	16.42	392.47	7.51
RegSeg [69]	54 px	84.98	77.53	81.08	48.44	78.44	59.89	119.55	8.40
ShorelineNet [3]	19 px	90.25	85.44	87.78	57.26	91.72	70.5	683.74	5.36
DeepLabV3_MBNV2 [12]_ [5]	29 px	90.95	80.62	85.47	82.54	86.84	84.64	308.42	14.71
ENet [41]	34 px	46.12	83.24	59.35	7.08	78.07	12.98	/	7.52
Full-BN [10]	33 px	71.79	85.34	77.98	20.43	84.53	32.91	649.96	11.37
TopFormer [29]	20 px	93.72	90.82	92.25	75.53	94.38	83.91	608.32	9.53
WaSR [1]	18 px	95.22	91.92	93.54	82.69	94.87	88.36	/	90.91
WaSR-Light (ours)	20 px	93.46	91.67	92.56	66.98	93.61	78.09	314.64	7.85
eWaSR (ours)	18 px	95.63	90.55	93.02	82.09	93.98	87.63	300.37	8.70

**Table 5 sensors-23-05386-t005:** Detection accuracy and latency on OAK-D and GPU of eWaSR using different encoders. We see that the standard eWaSR with ResNet-18 encoder (highlighted in blue) achieves the best F1 score overall and inside the danger zone. In the last column, we show the total number of channels in concatenated features that are passed through LSSE. The best results are denoted in bold.

		Overall			Danger Zone (< 15 m)			Latency		
bb	W-E	Pr	Re	F1	Pr	Re	F1	OAK	GPU	chs
MobileOne-S0 [18]	20 px	92.83	91.23	92.02	73.18	94.56	82.51	388.44	8.75	1456
RepVGG-A0 [17]	20 px	95.33	90.49	92.85	80.1	94.26	86.61	374.95	9.92	1616
RegNetX [15]	21 px	94.64	89.2	91.84	72.08	94.01	81.6	283.80	9.69	1152
MobileNetV2 [12]	21 px	93.44	88.35	90.82	68.91	92.96	79.15	193.90	8.25	472
GhostNet [13]	23 px	88.96	86.4	87.66	56.39	91.41	69.75	174.97	8.71	304
ResNet-18 [37]	18 px	95.63	90.55	93.02	82.09	93.98	87.63	300.37	8.70	960

**Table 6 sensors-23-05386-t006:** Results of ablation studies. We compare our proposed eWaSR (highlighted in blue) to modified eWaSR architecture without SRM on the penultimate SIM connection (¬long-skip), without SRM in LSSE block (¬SRM), and eWaSR with feature projection (reduction) on intermediate encoder outputs. We report overall and danger-zone F1 on MODS [46] benchmark and latency on OAK-D and GPU. The best results for each metric are denoted in bold.

		Overall	Danger	Latency	
Modification	W-E	F1	F1	OAK	GPU
eWaSR	18 px	93.02	87.63	300.37	8.70
¬long-skip	21 px	92.94	83.76	300.24	8.68
¬SRM	19 px	92.93	84.74	300.72	8.75
reduction	18 px	92.39	85.81	262.23	8.31

## Data Availability

eWaSR: Code available at https://github.com/tersekmatija/eWaSR (accessed on 15 April 2023); MaSTr1325: Publicly available at https://www.vicos.si/resources/mastr1325/ (accessed on 22 January 2023); MODS: Publicly available at https://github.com/bborja/mods_evaluation (accessed on 22 January 2023).

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
