# Peer review of "eWaSR—An Embedded-Compute-Ready Maritime Obstacle Detection Network"

_sensors, 2023, doi:10.3390/s23125386_

Round 1

Reviewer 1 Report

The manuscript designs an embedded-compute-ready maritime obstacle detection network. But the organization and representation of the manuscript make it confusing. Please find the detailed comments as follows:

1.       The authors have provided Fig. 2 and Fig. 3 regarding the architecture and methodological innovation of this work but authors should try to put more information, particularly in terms of dimensions and architectural parameters which will enhance the readability of the proposed work.

2.       Authors should try to explain all the terms given in Figures whether in their caption or text related to those figures.

3.       Table 1 needs more details, just putting a negative sign is not understandable. It can confuse.

4.       Similarly in Table 2: it is not clear why the Total execution time [ms] is less than the Max execution time [ms]. It needs to be properly explained.

5.       The manuscript is all about an embedded-compute-ready maritime obstacle detection network but there is no linkage of the proposed methodology to the maritime environment. The authors fail to show or discuss the context of maritime to design the proposed system.

6.       The authors did not discuss the problem of occlusion, illumination and other images related challenges which can be critical to assess the performance of the trained model.

7.       Authors should discuss the limitations of their work where the trained model find it difficult to predict the segmented mask which can make the way for possible future works.

8.       The acronyms should be defined before their use.

9.       The language of the manuscript can be improved. Authors should take care of typos mistakes and grammatical errors.

10.   The literature review should emphasize the limitations of the existing works and make the background which can motivate the proposed work. The stated limitations are not much convincing.

No major edit required. The language of the manuscript can be improved. Authors should take care of typos mistakes and grammatical errors.

Reviewer 2 Report

The paper proposes an embedded-compute-ready variant of WaSR (image segmentation problem). The authors replace computational intensive layers of WaSR to obtain the proposal eWaSR. 

As one of the main goals of this problem is to detect danger zone, I suggest the authors look at 3d object detection literature. I believe that this application is more important to estimate the distance of objects than the segmentation of the image. Also, discuss why you choose to perform segmentation. You can also develop a multitask model to achieve both distance and segmentation. 

Although you have Fig5, I suggest adding a graph based on Table 4 info, with F1 (y-axis) vs. Latency (x-axis), and mentioning how good your proposal is against the competitors.  With this figure, you can better convince the reader about your proposal. 

The proposal uses just a train and validation set. It would be nice also to have a test set. It might be possible that your results are overestimated, and the test set can help to identify these problems.

Overall is ok

Reviewer 3 Report

The content of the article is consistent with the scientific area of the journal Sensors . The subject raised by the authors is current and so far rarely noticed by other authors publishing in this area.
The described issues may contribute to increasing the efficiency of automation of the transport process in the Navy and reducing the risk of collision conditions in the future.
The paper is of an original scientific nature, which is related to an embedded computer network for the detection of marine obstacles. In this paper, we analyze the currently most powerful maritime obstacle detection network WaSR. Based on this analysis, the authors propose replacements for the most computationally intensive phases and we propose its embedded computer-ready variant eWaSR.
For a better clarification, please edit your paper as follows:
1. Expand the text of the manuscript (or the introduction or conclusion) with specific results in the world and in Europe, - increase the quality of the work by listing the results of publications of researchers and experts working in this field registered in world databases - wos. These are: Machine Vision System Measuring the Trajectory of Upper Limb Motion Applying the Matlab Software, Measurement of industrial robot pose repeatability, Visual Product Inspection Based on Deep Learning Methods, Contribution to Modal and Spectral Interval Finite Element Analysis or Registration of Holographic Images Based on Integral Transformation, thanks.
2. figures 4  and 6 should be contrasting and readable,
3. conclusions and future work should be extended to contain practical applications based on research described in this paper - expand references,
4. highlight the course of dependencies/relations in figure No. 2,  - the yellow color is indistinct,
5. unify the font in the tables.
I recommend publishing the post after the proposed modifications.

Round 2

Reviewer 1 Report

After conducting a comprehensive review of the manuscript and the author's responses, I am pleased with the modifications that have been made. The manuscript is well-written, lucid, and pertinent to the research question, with notable findings. Considering my evaluation, I endorse its publication. I commend the author for their hard work in refining the manuscript, and I believe it will add value to the current literature.

English is fine throughout the manuscript.

Reviewer 2 Report

The authors answered all my concerns 

Reviewer 3 Report

The authors accepted the comments, I recommend the paper to be published.